# EconAI: Preference-driven Agents Simulating Economic Activities via Large Language Model

## Abstract

The emergence of artificial intelligence has transformed the methodological frameworks in economic research by simulating intricate interactions among diverse agents. Despite the advantage of large language models (LLMs), they often struggle with occasions involving decision-making interactions with environments. This challenge stems from the fact that most LLMs are rationality-driven, seeking optimal economic benefits, while humans are preference-driven, pursuing the balance of personal goals (*e.g.,* income and health). These differences hinder the LLMs' ability to effectively understand economic activities across various contexts, leading to biases in economic simulations. To tackle this issue, we introduce **EconAI**, a novel approach aimed at enhancing the preference learning capabilities of LLMs by incorporating human-like preferences and cognitive processes. Specifically, EconAI features a 'knowledge brain' constructed from historical data and learning algorithms, enabling memory and making decisions for sophisticated economic facts. By integrating elements of self-learning, reflection, and experience updates, we refine decision-making processes, resulting in more accurate economic planning and mitigating planning bias in economic activities. Through the integration of real-time economic data and historical trends, EconAI offers a robust simulation platform that can adapt to market fluctuations and economic shocks. Our findings demonstrate that EconAI can model economic phenomena like inflation and employment with greater precision, showcase a notable ability to adjust to changing economic conditions, and surpass existing frameworks significantly.

## 1 Introduction

> "*Humans are not perfect optimizers. Instead, they seek satisfactory solutions rather than the optimal ones.*"
>
> — Herbert A. Simon

The advent of artificial intelligence (AI) has not only revolutionized methodological approaches in conventional economic research Jorgenson (2001), but it has also ushered in a new era of economic analysis. This paradigm shift is driven by the transformative impact of AI on data processing and pattern recognition, enabling economists to uncover intricate relationships and subtleties in economic data that were previously obscured or considered too complex to analyze Schorfheide & Song (2015); Christiano et al. (2005), facilitating economists with an unprecedented depth of insight into individual behaviors, consumer preferences, and market dynamics. In this way, conducting economic studies with AI becomes a promising direction.

Over the past two decades, agent-based modeling (ABM) has significantly evolved as a powerful framework for bottom-up simulations of economic systems, facilitating interactions among diverse agents without the constraints of a predetermined equilibrium Farmer & Foley (2009). This evolution can be primarily characterized by two distinct phases. Initially, ABM relied heavily on models with preset rules, which often incorporated overly simplistic assumptions about agent behaviors and interactions Tesfatsion & Judd (2006); Brock & Hommes (1998). The subsequent phase witnessed the emergence of learning-based models, which leveraged extensive behavioral data to more accurately

reflect complex economic dynamics Trott et al. (2021); Zheng et al. (2022); Mi et al. (2023). Despite the advances in agent-based modeling, tailoring decision-making processes to individual agents remains a complex challenge. Customized rule sets necessitate deep expert insight and intricate calibration efforts Windrum et al. (2007), whereas the use of specialized neural networks often results in exponentially increased computational demands and training complexities Mi et al. (2023). It impedes the practical application of such models, and also limits the ability to capture the rich diversity of economic dynamics in agent-based simulations.

Currently, the emergence of LLMs significantly improves agents' reasoning and planning skills, sparking a surge in new research Zhao et al. (2023b). However, if we directly apply LLM to tackle economic issues, they tend to be rationality-driven and cannot mimic human economic activities effectively Yue et al. (2024). As shown in Figure 1, there are differences in decision-making between LLM and humans, where these LLM-driven agents might aim for a single rational goal (*e.g.*, optimal economic benefit), resulting in choices that conflict with personal practices and essential preference criteria. In contrast, in reality, people are typically preference-driven and primarily make decisions based on their personal custom goals (*e.g.*, income and health), rather than economic rationality at most times, as confirmed by various economic studies Falk et al. (2018); Burks et al.

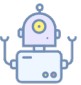
At this stage, banks have cut interest rates. What is your economic decision?

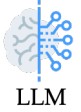
LLM
Based on my **economic knowledge**, it is advisable to prioritize investing over saving given the current financial conditions.

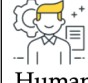
Human
Investing might be wise, but saving suits **my comfort with risk and long-term goals** in these uncertain times.

Figure 1: **The illustration of differences in decision-making between LLM and humans**. Humans are preference-driven while LLM are rationality-driven. It motivates us to develop the preference-driven LLM for economic simulation.

(2009). From the consideration above, this paper focuses on the following question:

*Can we develop agents that are preference-driven to simulate economic environments similarly to how humans do?*

To this end, we propose **EconAI**, a preference-driven agent with human-like characteristics for economic simulations. To refine our analysis, we focus on representative agents: households for microeconomic analysis and firms for macroeconomic perspectives. To enhance the realism of our economic simulations, we incorporate the influences of government and financial institutions, acknowledging their potential impacts on both macroeconomic conditions and the broader economic environment. Specifically, EconAI is equipped with a 'knowledge brain' for each type of agent—households and firms—built from their historical actions and learned knowledge by LLM. To achieve the precise modeling of agent preferences in decision-making processes, we propose three techniques: (1) self-learning from the observation, (2) self-reflection from the experience, and (3) self-updating for the preference and plan, to elicit helpful information from the interaction experience. In this way, it can model the influence of dynamic economic trends with EconAI, allowing agents to reflect on past experiences and market dynamics. In our experiments, traditional economic indicators such as market inflation and unemployment rates are simulated more accurately using our approach compared to conventional rule-based or machine-learning agents.

In summary, our contributions are three-fold:

- We recognize the flaw of the previously rationality-driven paradigm in economic decision-making, and pioneer the study of preference-driven agents. To our knowledge, we are the first to propose the preference-driven LLM, EconAI, designed to simulate economic environments in a manner akin to human behavior and thought.

- Inspired by the human learning process, we propose a new preference learning for LLM including self-learning, reflection, and updating modules to assimilate historical economic action data into our models. It can effectively model the preference, and provide interpretations for the thought and action process of humans in economic activities.

- We conduct macroeconomic and microeconomic simulations in our constructed environment driven by EconAI, and the performance surpasses other methods significantly. We observe various

economic behaviors from LLM-based agents that align with existing sociological and economic theories, informing future research and design implications.

## 2 RELATED WORK

### 2.1 SIMULATION IN MACROECONOMICS

In recent years, Agent-Based Modeling (ABM) has demonstrated superior potential in the realm of macroeconomic inquiry, outperforming traditional empirical statistical approaches Hendry & Richard (1982); Phelps (1967); Kydland & Prescott (1982) and Dynamic Stochastic General Equilibrium (DSGE) frameworks Christiano et al. (2005). In ABM, a multitude of autonomous agents engage in interactions predicated on established protocols or algorithmic constructs, thus circumventing the necessity for an a priori economic equilibrium hypothesis. Such an approach facilitates the exploration of a broad spectrum of non-linear dynamics, which is invaluable for policymakers seeking to conduct simulations of various policy interventions and to qualitatively evaluate their prospective economic repercussions.

However, agent-based models that employ fixed rules Tesfatsion & Judd (2006); Brock & Hommes (1998) or neural networks Trott et al. (2021); Zheng et al. (2022); Mi et al. (2023) have some draw backs. They are often criticized for their reliance on overly simplistic agent behaviors or an overreliance on extensive datasets for training, which can restrict their capacity to fully encapsulate the intricacies of economic dynamics. In our research, we present EconAI, an innovative model endowed with cognitive and strategic faculties. It is designed to emulate both macroeconomic and microeconomic phenomena in an adaptive manner, leveraging knowledge to enhance its predictive and analytical capabilities.

### 2.2 LLM-EMPOWERED AGENTS

LLMs, trained on vast corpora, have recently achieved human-like performance, laying the groundwork for sophisticated simulation agents Wang et al. (2023); Xi et al. (2023). These agents excel in simulation due to their autonomous adaptability Team (2022); Yoheinakajima (2023), strategic planning akin to human intelligence Wang et al. (2023); Xi et al. (2023), and their capacity for interaction with both agents and humans Park et al. (2023); Gilbert & Troitzsch (2005); Park et al. (2023). Their application has expanded into various fields, including social Park et al. (2022; 2023); Kovač et al. (2023); Gao et al. (2023); Jinxin et al. (2023) and natural sciences Boiko et al. (2023); Bran et al. (2023). In economics, they have been applied at three levels: individual behavior Horton (2023); Chen et al. (2023b), interactive planning and cooperation Guo (2023); Akata et al. (2023), and systemic market simulation Zhao et al. (2023a); Anonymous (2024); Chen et al. (2023a).

However, current research such as Li et al. (2024) is mostly rationality-driven and has yet to explore preference learning within a multi-agent environment in a manner that reflects human-like decision-making processes. Our work addresses this gap by focusing on the preference-driven agents for simulation.

## 3 PRELIMINARY

This section outlines our economic simulation's framework, depicted in Figure 2. Adhering to established simulation methodologies, our model integrates the EconAI to drive the environment, focusing on four main areas: household, firm, financial institution, and government, which can form an economic system including both *macroeconomic* and *microeconomic* environments. The simulation models key real-life decisions—working and consuming—as pivotal economic activities Gatti et al. (2011); Wolf et al. (2013); Dawid & Gatti (2018), which, in turn, affect government tax income Zheng et al. (2022); Trott et al. (2021); Dawid & Gatti (2018) and the behavior of the labor and consumer markets Lengnick (2013); Deissenberg et al. (2008); Dawid et al. (2012). Based on these market conditions, banks modify interest rates to align with inflation or deflation trends Wolf et al. (2013); Dawid & Gatti (2018).

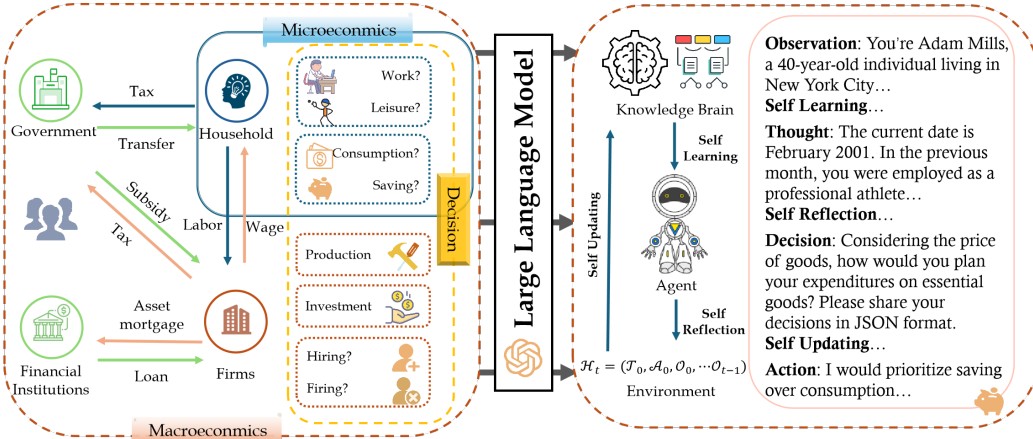

Figure 2: The illustration of the simulation for the microeconomic and macroeconomic environment (left) and our EconAI (right). On the left, the microeconomic decisions of households regarding work and leisure are analyzed, while the macroeconomic decisions of firms concerning production, investment, and employment are displayed. These decisions are influenced by simplified interactions with government and financial institutions. On the right, the EconAI involves preference learning including self-learning, reflection, and updates within an LLM that observes and interacts with the environment. In this way, it can inform economic activities and make human-like decisions.

**Background.** Language agents primarily interact with the world by generating internal thoughts and actionable outputs. This study builds upon and advances the action trajectory framework introduced in Yao et al. (2023a). A typical planning trajectory, $\mathcal{H}$, involves a sequence of Thought-Action-Observation $(\mathcal{T}, \mathcal{A}, \mathcal{O})$, where $\mathcal{T}$ encapsulates the agent's internal thoughts, $\mathcal{A}$ denotes the actions taken, and $\mathcal{O}$ captures the environmental feedback. The historical context $\mathcal{H}$ up to a time point $t$ is expressed as follows:

$$\mathcal{H}_t = (\mathcal{T}_0, \mathcal{A}_0, \mathcal{O}_0, \mathcal{T}_1, ..., \mathcal{T}_{t-1}, \mathcal{A}_{t-1}, \mathcal{O}_{t-1}) \tag{1}$$

Based on this historical data, the agent generates thoughts $\mathcal{T}_t$ and actions $\mathcal{A}_t$. The generation process for the next thought, given by the language model $\pi$ with parameters $\theta$, is mathematically modeled as follows:

$$p(\mathcal{T}_t|\mathcal{H}_t) = \prod_{i=1}^{|\mathcal{T}_t|} \pi_\theta(\mathcal{T}_t^i|\mathcal{H}_t, \mathcal{T}_t^{<i}), \tag{2}$$

where each token $\mathcal{T}_t^i$ and the total length $|\mathcal{T}_t|$ are considered. Following thought generation, the corresponding action $\mathcal{A}_t$ is determined:

$$p(\mathcal{A}_t|\mathcal{H}_t, \mathcal{T}_t) = \prod_{j=1}^{|\mathcal{A}_t|} \pi_\theta(\mathcal{A}_t^j|\mathcal{H}_t, \mathcal{T}_t, \mathcal{A}_t^{<j}), \tag{3}$$

Here, $\mathcal{A}_t^j$ refers to the j-th token and $|\mathcal{A}_t|$ to the length of the action sequence. The outcomes of these actions are then observed as $\mathcal{O}_t$, contributing to the next iteration of the trajectory, $\mathcal{H}_{t+1}$. Notably, the actions $\mathcal{A}_i$ within the trajectory are explicitly equivalent to action $a_i$ in the later discussion regarding the action set $E_a$

## 4 KNOWLEDGE-ADAPTIVE AGENT FOR ECONOMICS

In this section, we propose EconAI, which treats $\mathcal{X}$ as the decision-making for an economic activities plan that includes a sequence of abstract actions to execute in different scenarios. Economists Falk et al. (2018) propose using preferences as the cause for decision-making for participants in economic

| Prompt Name | Prompt Content |
|---|---|
| Thought-prompt | Identify which step of the plan you are at. Show your thoughts about the one next action. Your thoughts should be faithful to the plan step. |
| Summary-prompt | Summarize the interaction history in steps, and think about the flaws in the previous experience. |
| Forward-looking -prompt | Look ahead to your future life, and think about what you should do in next. |
| Preference-prompt | Evaluate the satisfaction with your current life, and think about the next plan. |
| Upd-*-prompt | Based on the above experiences and thoughts, Update your knowledge about *. |

Table 1: Prompts that EconAI uses in Economic environment.

activities. To this end, we utilize the action trajectory to model and capture the preferences. As shown in Figure 2, we design a four-stage process to optimize plan $\mathcal{X}$ iteratively: 1) leverage the willingness and utility to model the preference through a knowledge brain, 2) self-learning with the current action trajectory, 3) self-reflection on the collected experiences, and 4) self-updating for the decision-making plan and knowledge brain.

**Problem Setting.** We aim to design an LLM-based agent to accomplish an economic activities modeling problem. The agent is provided with a natural language description of the task, possible actions, and environmental observations. Let $\mathcal{M}$ be the LLM agent, $\mathcal{A}$ be the set of possible actions, and $\mathcal{O}$ be the set of possible observations from the environment. One could augment the input with a custom economic plan $\mathcal{X}$. At each step $t$, the agent $\mathcal{M}$ generates a text action $a_t \in \mathcal{A}$ and receives a text observation $o_t \in \mathcal{O}$ from the environment. $o_0$ denotes the initial observation, which could be empty. We define a preference module $\mathcal{P}(o_{0:t})$ related to some indicators such as income, satisfaction, and health. Our goal is to design an optimal economic plan $\mathcal{X}$ to maximize the expected preference over all possible task instances,

$$\mathcal{X}^* = \arg\max_{\mathcal{X}} \mathbb{E}_P \left[ \mathcal{P}(o_{0:T}) \right], \tag{4}$$

where $T$ is the maximum number of interaction steps allowed.

## 4.1 RULE-RELEVANT KNOWLEDGE BASE

For an agent, such as a household, it has specified metadata such as the profession, specialty, skills, credentials, and experiences of the agent. The agent observes information from the environment, makes decisions, and conducts the appropriate action. In real-world economic activities, humans often make decisions with the assistance of their experimental rules and customs, such as the decline in bank interest rates is conducive to investment. Afterward, people reuse these rules of thumb based on their successes or update their own rules of thumb based on their failures on specific occasions. Much like humans, the agent's brain serves as a central nucleus driven by an LLM. The brain module enables the agent to exhibit sophisticated cognitive abilities critical for professional-grade performance, including memory, planning, and reasoning. To mimic this vital component for the agent, we design a knowledge brain as follows.

**Action and Rules.** The action set $\mathcal{E}_a = \{a_1, ..., a_{N-1}\}$ encompasses a collection of discrete actions that LLMs must execute to perform specific functions effectively. The rule set $\mathcal{R} = \{r_1, ..., r_{N-1}\}$ then defines the logical order and conditions for action transitions within the system, such as "If I have enough savings, I turn my focus to invest." These rules are essential for guiding allowable transitions $r_k : a_i \rightarrow a_j$, which are determined by the inherent linkages among actions or by specific task requirements.

**Knowledge Brain.** Action knowledge, expressed as $(\mathcal{E}_a, \mathcal{R})$, includes both a structured set of actions $\mathcal{E}_a$ and the corresponding rules $\mathcal{R}$ that govern their sequencing. This collective body of knowledge, referred to as the *Knowledge Brain*, integrates action sequences for various tasks and provides critical support for action generation and decision-making processes. Given the vast and varied action knowledge required for numerous tasks, creating this entirely manually is impractical

and labor-intensive. To overcome this, and to leverage the robust capabilities of LLMs demonstrated in related tasks (Liu et al., 2023), we first employ GPT-4 (OpenAI, 2023) for preliminary construction, which is then finely tuned through manual refinement.

## 4.2 SELF LEARNING

With the knowledge brain defined above, we can leverage it to model the preferences of the agent in economic activities. Additionally, it facilitates the agent's thinking process in decision-making. As ReAct Yao et al. (2023b) mentions, a "thought" action does not elicit any environmental feedback and solely reflects the reasoning process of the LLM.

In this way, EconAI starts with an empty plan $\mathcal{X}_0$. At each iteration $t$, each agent makes decisions based on the knowledge brain $\mathcal{B}$ and the previous action history. For each household or firm, the LLM agent generates a sequence of thoughts and actions in response to observations from the environment:

$$\mathcal{H}_{t-1} = \mathcal{X}_{t-1} \oplus (o_0, \tau_0, a_0, o_1, \cdots, o_{t-1}).$$

where $\oplus$ means combining together in the same sequence. Since we augment the action space with thoughts that do not affect on the environment, at each step $t$, EconAI first obtains the thought,

$$\tau_t = \mathcal{M}(\mathcal{H}_{t-1} \oplus \text{Thought-prompt}) \tag{5}$$

where Thought-prompt is provided to make the LLM agent act faithfully to the plan $\mathcal{X}_i$. Then we sample the next action given the thought $\tau_t$,

$$a_t = \mathcal{M}(\mathcal{H}_{t-1} \oplus \tau_t \oplus \mathcal{B}_{t-1}) \tag{6}$$

$$\mathcal{H}_t = \mathcal{H}_{t-1} \oplus \tau_t \oplus a_t \oplus o_t. \tag{7}$$

where $o_t$ is the observation after action $a_t$.

## 4.3 SELF REFLECTION

Humans tend to apply a kind of heuristic thinking to reflect the complex task and then summarize this activity as an experience. Therefore, the reflection component of the EconAI brain is designed to perform as humans when faced with an elaborate task as follows.

Given the experience $\mathcal{H}_T$ and the corresponding preference $\mathcal{P}(o_{0:t})$ (denoted as $\mathcal{P}_t$), we instruct the LLM agent to reflect on the interaction history through a self-reflection procedure for this interaction history:

$$s_t = \mathcal{M}(\mathcal{H}_t \oplus \mathcal{P}_{t-1} \oplus \mathcal{B}_{t-1} \oplus \text{Summary-prompt}) \tag{8}$$

$$f_t = \mathcal{M}(\mathcal{H}_t \oplus \mathcal{P}_{t-1} \oplus \mathcal{B}_{t-1} \oplus \text{Forward-looking-prompt}) \tag{9}$$

$$p_t = \mathcal{M}(\mathcal{H}_t \oplus \mathcal{P}_{t-1} \oplus \mathcal{B}_{t-1} \oplus \text{Preference-prompt}) \tag{10}$$

where Summary/Forward-looking/Preference-prompts are shown in Table 1. $\mathcal{P}_t$ can be evaluated by LLM by inputting the previous decision, action, and the current state of the agent.

## 4.4 SELF UPDATE

The human being can store and update the knowledge learned from the real world, *e.g.*, observations, thoughts, and actions. Similar to the processes of human strategy formulation, the knowledge of agents also should update the useful information and adapt to the new occasion.

With the knowledge base $\mathcal{B}_t$, the current task plan $\mathcal{X}_t$, $s_t$, $f_t$, and $p_t$, we utilize the LLM to revise $\mathcal{X}_{t-1}$ and obtain an improved plan $\mathcal{X}_t$ and update the knowledge base $\mathcal{B}_t$ as follows:

$$\mathcal{X}_t = \mathcal{M}(\mathcal{X}_{t-1} \oplus \mathcal{B}_{t-1} \oplus (s_t, f_t, p_t) \oplus \text{Upd-}\mathcal{X}\text{-prompt}) \tag{11}$$

$$\mathcal{B}_t = \mathcal{M}(\mathcal{X}_{t-1} \oplus \mathcal{B}_{t-1} \oplus (s_t, f_t, p_t) \oplus \text{Upd-}\mathcal{B}\text{-prompt}) \tag{12}$$

where Upd-*-prompt asks the LLM to generate an updated version for $* = \mathcal{X}$ or $\mathcal{B}$, given the task instances and reflections. After obtaining a revised plan $\mathcal{X}_{i+1}$, we continue the iterative process until we reach maximum optimization iterations $T$. During inference, we follow the same procedure as experience collection except that now we use the final optimized plan $\mathcal{X}_T$.

In summary, EconAI initially models agents' preferences using knowledge brains and then utilizes the self-learning, reflection, and updating process for interactive decision-making in economic activities. This approach establishes a human-like simulation within the economic environment, which will be further evaluated for performance in the subsequent section.

## 5 EXPERIMENTS

In this section, we conduct experiments to study the ability of EconAI, aiming to answer the following research questions (RQ).

- **RQ1**: How does the EconAI behave in simulation, compared with the traditional models?
- **RQ2**: How do the main components in EconAI affect the simulation results?
- **RQ3**: Does the decision-making mechanism of EconAI possess interpretability, and can the simulation based on EconAI reflect the impact of external intervention?

### 5.1 EXPERIMENTAL SETUP

**Baselines.** We select **LEN** Lengnick (2013) and **CATS** Gatti et al. (2011) as baselines because 1) they partially reproduce the aforementioned macroeconomic phenomena within their own (more complex) simulation frameworks, and 2) their carefully designed decision rules for work and consumption are representative, reflecting typical decision-making observed in real-life scenarios. Given the importance of agents' heterogeneity in macroeconomic simulations, we also combine these two baselines into an additional baseline, **Composite**, where each agent randomly adopts one of the decision rules. In addition, we select a learning-based method, AI-Economist Zheng et al. (2022) (**AI-Eco**), which builds on the assumption of rational decision-making and employs reinforcement learning (RL) Arulkumaran et al. (2017) to maximize the agent's utility. Moreover, we compare our approach with **EconAgent** Li et al. (2024), which includes a perception module to model the macroeconomic environment and creates heterogeneous agents with distinct decision-making mechanisms.

**Definition of Economic Indicators.** Annual nominal GDP is defined as the sum of $S \times P$ over one year. As for real GDP, we set the first year in the simulation as the reference year and replace $P$ with $P_0$, where $P_0$ is the goods price in the reference year. The definition of the annual (price) inflation rate and the unemployment rate is shown in Eq. 25. For wage inflation, the definition is similar to that of price inflation, where the average price is replaced with the average wage across all the agents. For households, disposable income is defined as the total income after taxes and essential expenditures. The savings rate is defined as the proportion of disposable income that is saved rather than spent on consumption. For firms, profit margin is defined as the ratio of net profit to total revenue, indicating the profitability of the firm.

**Simulation Setup.** In an effort to exploit the comprehensive understanding and contextual knowledge of Large Language Models (LLM), each simulated agent is equipped with distinct real-life attributes such as name, age, and occupation. The LLM autonomously generates names that are then randomly allocated to each agent. The age profile for the agents adheres to the demographic distribution of the U.S. population between ages 18 and 60, as reported in 2018 Bureau (2024). Regarding economic variables, the simulation adjusts the scale parameters of the Pareto distribution for hourly wages to ensure that the synthesized monthly wages correspond with actual U.S. economic figures and taxation categories from 2018 Zheng et al. (2022). Additionally, the LLM is tasked with creating ten distinct job titles for each decile of this wage distribution, reflecting the substantial wage variances observed across different employment types in real life. Job assignments are dynamically regulated: agents retain their jobs if employed in the previous month or receive a new job offer, determined by the prevailing wage distribution, if previously unemployed. Details on the age and wage distributions as well as job classifications are included in the supplementary materials. The simulation framework was developed using Python, leveraging the capabilities of GPT-3.5-turbo-0613 provided through the OpenAI API[1].

---

[1] https://platform.openai.com/

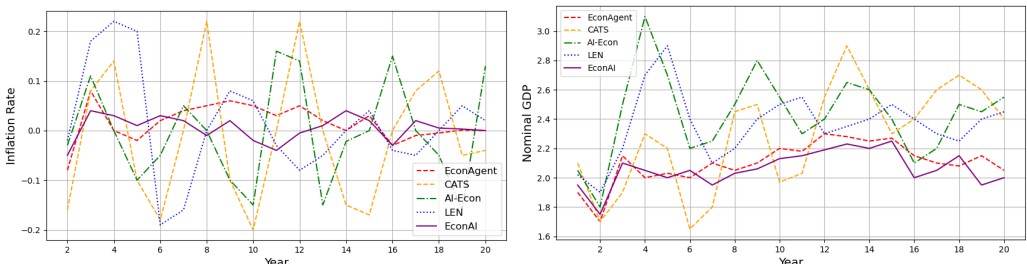

Figure 3: Annual variations of macroeconomic indicators, where the simulation based on EconAI shows more stable and numerically plausible indicators.

## 5.2 MACRO-LEVEL ANALYSIS (RQ1)

**Economic Indicators.** In Figure 3, we depict the fluctuations of the annual inflation rate and nominal GDP. Note that the unreasonable unemployment rate (around 46%) and nominal GDP for AI-Eco are not reported. Both rule-based and RL-driven baselines produce **anomalous indicators and large fluctuations**. In contrast, agent decision-making based on EconAI has demonstrated more **stable and numerically plausible** macroeconomic phenomena across multiple dimensions, even without fine-tuned calibration. This suggests that EconAI's decision-making is coherent and more closely emulates real-world human behavior, leading to a more natural equilibrium between supply and demand in the consumption market. We also

Table 2: Prediction error of different models. Up, SR, and PM denotes unemployment, saving rate, and profit margin, respectively.

| Model | Inflation | Up | SR | PM |
|---|---|---|---|---|
| LEN | 0.325 | 0.265 | 0.257 | 0.344 |
| CATS | 0.304 | 0.218 | 0.187 | 0.266 |
| Composite | 0.255 | 0.176 | 0.149 | 0.203 |
| AI-Eco | 0.355 | 0.294 | 0.206 | 0.285 |
| Econ-Agent | 0.197 | 0.134 | 0.153 | 0.168 |
| EconAI | **0.146** | **0.112** | **0.139** | **0.127** |

compare our EconAI with other baselines for the prediction error, which can be measured by the mean square error of the forecast values for each year compared with the true facts. As shown in Table 2, EconAI can achieve the best results, demonstrating its reasonability and effectiveness.

**Economic Regularity.** As one of the most commonly used regularities in macroeconomic simulations for validating the plausibility of simulation results, the Phillips Curve Phelps (1967) describes the negative correlations between the annual unemployment rate and wage inflation. As shown in Figure 4, only the decision-making of EconAI has **correctly manifested phenomena in accordance with these two regularities** (Pearson correlation coefficient is -0.522, $p < 0.01$). Notably, the rule-based baseline method displayed an **incorrect positive relationship on the Phillips Curve**. We attribute this advantage to the EconAI's accurate perception that consumption should be reduced when unemployed.

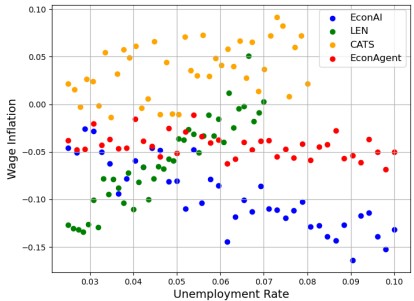

Figure 4: Economic regularity study.

## 5.3 MICRO-LEVEL ANALYSIS (RQ1)

In the economic environment established by EconAI, we can observe that there can reveal the classic market strategies including *differentiation*, *imitation*, and *customer orientation*.

**Differentiation.** Differentiation is a generic strategy that allows competitors to occupy a unique market position Porter (1997). Approaches to differentiation can take many forms: design brand image, customer service, or other dimensions. These approaches can also be observed in our environment. The following is a clip showing a competitor trying to focus on signature products to establish its own brand:

Table 3: Interactions between Households and Firms in an Economic System

| Household Need | Household Behavior | Type |
|---|---|---|
| Employment Stability | Pursue higher education and training | Employment Policy |
| Retirement Savings | Opt into firm-provided retirement plans | Financial Planning |
| Work-Life Balance | Demands to balance life and work | Work Environment |
| **Firm Need** | **Firm Behavior** | **Type** |
| Skilled Workforce | Offer long-term contracts | Employment Policy |
| Secure Long-term Employees | Provide matched retirement saving plans | Financial Planning |
| Increase Productivity | Flexible working hours and remote work | Work Environment |

*Expand the direction to exploit the latent products that can become customer favorites and differentiate us from our competitors (Based on product differentiation and market segmentation).*

*Streamline the direction to focus on a few high-quality, signature products that can become customer favorites and differentiate us from our competitors (Focused on economies of scale and brand strengthening).*

**Imitation.** Imitation is also a classic strategy that actively observes and adapts to the strategies of its competitors to maintain competitive parity or limit rivalry in market competition (Lieberman & Asaba, 2006). The following is another clip showing how another competitor finds its rival advantage and decides to imitate.

*The new product may meet risk. I will not study and develop new products at this time (Incorporates risk aversion and precautionary principle in uncertain economic conditions).*

*The new product is a clear advantage. I will study and develop the new products (Reflects opportunity cost and potential for higher returns in a favorable economic environment).*

**Agent Orientation.** Firms discover and cater to labor needs to help them gain advantages in competition Zeithaml et al. (2018). Those who prioritize labor insights are better positioned to adapt, innovate, and thrive amidst competition. 3 shows the agent responses tailored to different needs. Notably, agents can not only identify individual needs but also assess trends in factors (*e.g.*, employment policy), allowing them to make adjustments accordingly.

## 5.4 ABLATION STUDY (RQ2)

We separately remove the perception module and the reflection module, and the results of 10 years are as shown in Figure 5. We observe that when there is no perception capability, the inflation rate and unemployment rate fluctuations significantly decrease, appearing "too stable", especially for the unemployment rate. This suggests that the agents have low sensitivity to changes in their economic conditions and cannot make adaptive decision adjustments. When there is no reflection capability, the inflation rate exhibits anomalies close to 15% in the first three years, emphasizing the importance of long-term (a quarter in our experiments) economic environment perception.

## 5.5 EXTERNAL INTERVENTION (RQ3)

We extend to examine how external factors influence agent-based decisions, a critical aspect frequently explored in economic ABM literature Dawid & Gatti (2018). The COVID-19 pandemic serves as a pivotal example of such external shocks, given its profound effect on the world's economic landscapes. To simulate the effects of COVID-19 accurately, we embed related scenarios directly within EconAI's

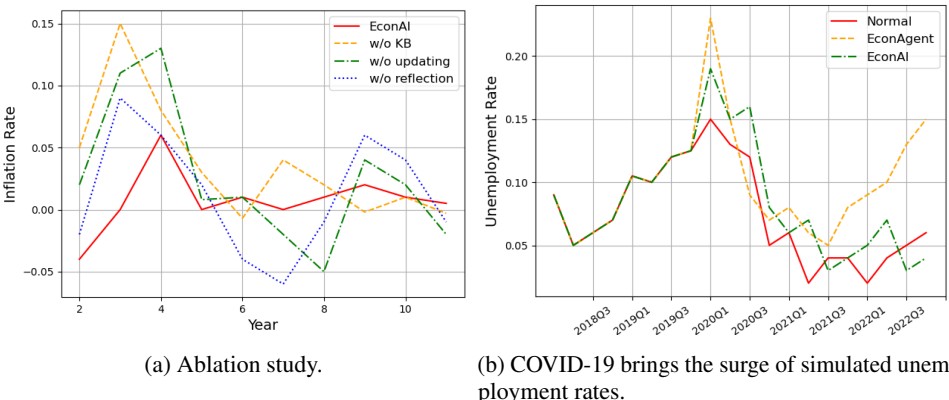

(a) Ablation study.

(b) COVID-19 brings the surge of simulated unemployment rates.

Figure 5: More experimental studies.

prompts. From March 2020 onwards, our simulations include a special directive to model its economic implications, as illustrated below:

> *Since March 2020, the outbreak of COVID-19 has led the U.S. federal government to declare a national emergency, reflecting a significant disruption across various economic sectors.*

**Analysis of Unemployment Trends.** As depicted in Figure 5, we present a comparative analysis of unemployment rates, labeled 'Normal' and 'COVID-19' to represent scenarios with and without the aforementioned prompt, respectively. The data illustrate that our EconAI model accurately reflects the spike in unemployment observed in the first quarter of 2020 due to the COVID-19 crisis Organization for Economic Co-operation and Development (1970). While the figures don't align precisely with actual statistics, they underscore the capability of our framework to qualitatively capture the essence of human decision-making and macroeconomic dynamics in authentic scenarios. Additionally, the persistent elevation in unemployment rates past 2021, without government intervention measures in our model, mirrors the prolonged repercussions of the pandemic observed in the 'COVID-19' scenario compared to the 'Normal' conditions.

The following is an example of the agent's reflection during COVID-19, demonstrating its human-like decisions and updating its experimental rules:

> *...(1) When economic uncertainty rises (e.g., job security declines), individuals should lean toward risk aversion, reducing work participation or seeking more stable employment. (2) Without government intervention, individuals should anticipate prolonged economic downturns and adjust expectations and activities with caution.*

## 6 CONCLUSION

In this work, we ventured into the novel integration of LLMs with macroeconomic simulation, designing EconAI with the abilities of self-learning, self-reflection, and self-update for decision-making based on the context of real-world economic environments. Our method involves utilizing action knowledge to guide the model's action generation, translating this knowledge into text for deeper model comprehension, and employing a *knowledgeable self-learning* phase for continuous improvement. EconAI can effectively model the classic macro/micro-economic phenomena that are reproduced and more reasonable compared to traditional rule-based or learning-based agents. Through this endeavor, it has become evident that the capabilities of LLMs offer a promising avenue to simulate more realistic economic activities.

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
