# OpenReview forum: "EconAI: Preference-driven Agents Simulating Economic Activities via Large Language Model"
_ICLR.cc/2025/Conference — ICLR 2025 Conference Withdrawn Submission_

### Official Review · Reviewer_XqCG · 2024-10-22

**Soundness:** 2
**Presentation:** 3
**Contribution:** 2
**Rating:** 3
**Confidence:** 4

**Summary:**

This paper builds on the work of EconAgent by using a multi-LLM agent framework to simulate economic societies. It incorporates self-learning, self-reflection, and self-updating prompting methods to enhance the decision-making agents (simulated by LLMs) with human-like preferences and cognitive processes.

**Strengths:**

Overall, the paper is well-written and easy to follow, and some of the experimental findings, such as the impact of COVID-19 on the simulated economic society, are particularly interesting.

**Weaknesses:**

I have the following concerns regarding this work:

- Limited technical innovation: Based on my understanding, the framework setup have largely been addressed in previous work [1]. In comparison, this paper's main contribution lies in incorporating preferences into the agent by defining agent profiles and enhancing prompting. While the proposed prompting methods—self-learning, self-reflection, and self-updating—seems effective, they appear to be built upon well-established techniques such as ReAct [2] and reflection [3], which might somewhat limit the novelty of the contribution.

- The author argues that the flaw in previous studies is assuming rational agents without human preferences. I think this isn't necessarily a flaw—it depends on the research objective. Rational agents are suitable for economic equilibrium models, but if the goal is to study complex social interactions or behavioral economics, agents should reflect human-like preferences, which involves two key aspects: 1. Agent profile distribution: The distribution of agent attributes should reflect real-world demographics and economic conditions. The author's approach here is quite solid. 2. Authenticity of agent decisions: The author overlooks whether agents' decisions reflect real human preferences. While challenging to verify, some targeted analysis of the LLM’s decision-making consistency would help.

- Since EconAgent [1] already analyzed rule-based and RL-based baselines, I expect EconAI to show clear improvements. However, the results are very similar, and there’s no detailed comparison or qualitative analysis between the two models.

[1] Li N, Gao C, Li M, et al. Econagent: large language model-empowered agents for simulating macroeconomic activities[C]//Proceedings of the 62nd Annual Meeting of the Association for Computational Linguistics (Volume 1: Long Papers). 2024: 15523-15536.
[2] Yao S, Zhao J, Yu D, et al. React: Synergizing reasoning and acting in language models[J]. arXiv preprint arXiv:2210.03629, 2022.
[3] Shinn N, Cassano F, Gopinath A, et al. Reflexion: Language agents with verbal reinforcement learning[J]. Advances in Neural Information Processing Systems, 2024, 36.

**Questions:**

- An evaluation of the consistency of the LLM’s decision-making, along with additional qualitative analysis, would provide further insights.
- I believe the experiment should include more comparisons between EconAgent and econAI, as the current results appear very similar.
- The citation format is incorrect; please use \cite or \citep appropriately.

---

### Official Review · Reviewer_6M76 · 2024-11-03

**Soundness:** 2
**Presentation:** 1
**Contribution:** 2
**Rating:** 3
**Confidence:** 4

**Summary:**

The paper introduces an LLM-enhanced agent-based simulation framework for modeling economic activity. Unlike previous approaches, the authors incorporate innovations such as self-reflection and updates to a knowledge base ("brain") to better align the LLM agent with human cognitive and decision-making processes. In their experiments, the authors compare the proposed method to prior frameworks, demonstrating that the simulation results align with the Phillips Curve. Specifically, the results show a negative correlation between simulated wage inflation and the unemployment rate. Additionally, the model responds realistically to an embedded external shock—specifically, the impact of COVID-19—during the simulation.

**Strengths:**

1. The paper addresses a timely topic and presents an intriguing application of LLM agents, leveraging them to simulate complex economic behaviors. Given that LLMs are increasingly aligned with human preferences and equipped with prior knowledge, this research direction has the potential to enable more realistic simulations and become a powerful tool for economic research.

2. The design concept for the LLM agent is a notable strength. The inclusion of self-reflection and self-updating knowledge abilities establishes a foundation for creating agents capable of assessing their trajectories and adapting over time, which aligns well with human-like decision-making processes.

**Weaknesses:**

1. One weakness of the paper is that the clarity of the presentation could be significantly improved.
  - Some of the concepts are introduced without providing concrete examples or detailed explanations, which leaves readers with an incomplete understanding of this core component, limiting the model's interpretability and practical applicability. For instance, the authors do not clarify what "an economic plan" entails in a concrete sense. It remains ambiguous whether the economic plan is a simple, modifiable prompt or a more sophisticated construct with a parametric structure that enables iterative updates. If the latter is true, it is unclear how this plan is actually updated—whether through specific parameter adjustments, feedback from the environment, or internal decision-making criteria.
  - The paper uses excessive and inconsistent notation, reducing clarity. For example, the action space is denoted as "A" on lines 238-239 but as "ε" on lines 259-260. A flow chart covering all design elements mentioned in the paper would improve readability, providing a clear overview.

2. The paper lacks sufficient implementation details in the method and experiment sections. While the framework’s primary outputs are economic variables, the authors do not explain how these variables are aggregated and measured. Although agent design is discussed, the calculation of outputs remains unclear. Additionally, the paper does not specify the input data required for the framework, including prompt templates and parameters for the LLM, making it challenging to reproduce the results. Providing these details would enhance transparency and reproducibility.

3. The authors do not provide an adequate evaluation of the framework’s authenticity, nor do they critically assess its limitations. Subjective evaluations without comparison to real-world scenarios should be avoided. For example, in the ablation study (Section 5.4), the framework’s “too steady” behavior is labeled unfavorably without objective criteria. Additionally, in Section 5.3, it is unclear how certain observed market strategies relate to enhanced economic simulation, as these strategies might already exist in the training corpus rather than emerging from the framework’s modeling capabilities.

**Questions:**

1. Could the authors provide an end-to-end example of the workflow and how the economic variables are aggregated from the framework?

2. Could the authors justify why the marketing strategy is related to economic simulation analysis?

**Minors:**
1. Lines 356-357 referenced equation 25, which does not exist in the paper.

---

### Official Review · Reviewer_mfqy · 2024-11-03

**Soundness:** 2
**Presentation:** 1
**Contribution:** 1
**Rating:** 3
**Confidence:** 4

**Summary:**

The paper proposes a framework that enables the LLM to have its own preferences and interact with the market environment.  EconAI incorporates human-like preferences and cognitive processes, including self-learning, reflection, and experience updates, to enhance its decision-making abilities.

**Strengths:**

(1) The framework is sound for learning from observations, reflecting on insights, and updating experiences.
(2) The paper attempts to model both macroeconomic and microeconomic perspectives, which is meaningful for simulating market environments. This environmental information is used to interact with agents and impact agent's action. Examining how the environment interacts with agents is a practical field of research.

**Weaknesses:**

The framework for learning, reflection, and self-updating may lack novelty. Guo, Taicheng, et al. "Large language model based multi-agents: A survey of progress and challenges." arXiv preprint arXiv:2402.01680 (2024). This paper summarized the Agents-Environment Interface as well as Economic simulation.

**Questions:**

(1) In the abstract, "most LLMs are rationality-driven, seeking optimal economic benefits ..." Could you explain the logic relationship or provide any literature reviews that support this claim? In my opinion,  LLMs itselves do not possess intentions, desires, or goals. They are tools designed to process and generate text based on patterns and data they have been trained on. They do not seek outcomes in the way humans or even economically rational agents might. Additionally, the use of "most" also implies there are some that are not.

(2) In line 051, the paper mentioned the ABM relied heavily on models with preset rules, which often incorporated overly simplistic assumptions about agent behaviors and interactions.  Following this, the paper mentioned the advances in agent-based modeling without specifying what these advances entail. Later, in line 057, the discussion introduces neural networks, yet it is unclear how these networks relate to the previously mentioned agent-based topics. Additionally, the specific tasks these neural networks are intended to perform within this context are not clearly defined.

(3) In figure 1, could you clarify which LLM generated this solution? When testing several LLMs, I've noticed that the results typically account for various situations across different scenarios or prompt me to provide more detailed background information or clarify my goals, rather than directly providing straightforward solutions.

(4) In Figure 2, can you explain how the agent's preferences interact with environmental information.

(5) In table 2, can you explain how did you use EconAI to predict inflaction, unemployment, saving rate and profit margin?

(6) In Figure 5, do you have statistical evidence to illustrate the performance across each scenario? Is the observed outperformance significant, given that the points on the curve are close.

---

### Official Review · Reviewer_hMWE · 2024-11-03

**Soundness:** 3
**Presentation:** 3
**Contribution:** 2
**Rating:** 5
**Confidence:** 3

**Summary:**

This paper introduces EconAI, an approach to simulate macroeconomic trends using LLM agents.

**Strengths:**

The EconAI framework attempts to model individual agent's cognitive process in learning and decision making. The idea of incorporating a knowledge brain in simulating agent behaviors is novel. This is useful for generating more realistic and fine-grained agent's behavior and could lead to more interesting macro-level group behaviors.

The paper is clearly written and easy to read. It includes both micro and macro analysis and an ablation study. Overall the evidence tend to support the effectiveness of the proposed method.

**Weaknesses:**

1. The paper is quite light on implementation details (how many agents are simulated? what hyperparameters did you use? etc.) and no code has been attached. It's unclear if the results will be reproducible.

2. The paper argues that the proposed method can take individual's preference into account when simulating the agents. However, it's unclear how these preferences are generated on an individual level (e.g. how certain agents are more risk averse than others). Or maybe the authors only interpreted "preference" to be rules/policies learnt from experiences. This needs to be clarified.

3. The micro-level analysis is quite shallow and only includes some qualitative responses from the LLM. To show how the knowledge brain can improve the simulation quality on a micro-level, the paper should compare the results against other models such as EconAgent. Including more micro-level quantitative measures would be useful, for example, what knowledge/preferences do different agents have/learn.

4. It would be great if the authors can include some example scenario prompts and model responses in the appendix.

5. The paper fails to discuss the limitation of the EconAgent model.

**Questions:**

1. The paper mentioned that the Knowledge Brian was constructed using GPT4 while the simulation is done in GPT3.5. Could you clarify how/why different models are used?

2. In Table 2, is the performance difference statistically significant? Why not plot the ground-truth economic indicators in Fig 3?

3. Is the EconAI model more costly than other LLM baselines such as EconAgent?

4. Have you performed any sensitivity analysis to show how parameters affect simulation results?

5. The paper mentioned in 4.1 that "we first employ GPT-4 (OpenAI, 2023) for preliminary construction, which is then finely tuned through manual refinement." Could you clarify what is meant by fine-tuning with manual refinement? What data did you use?

---

### Note · Authors · 2025-01-28

I have read and agree with the venue's withdrawal policy on behalf of myself and my co-authors.